# Distinct Impact of Natural Sugars from Fruit Juices and Added Sugars on Caloric Intake, Body Weight, Glycaemia, Oxidative Stress and Glycation in Diabetic Rats

**DOI:** 10.3390/nu13092956

**Published:** 2021-08-25

**Authors:** Tamaeh Monteiro-Alfredo, Beatriz Caramelo, Daniela Arbeláez, Andreia Amaro, Cátia Barra, Daniela Silva, Sara Oliveira, Raquel Seiça, Paulo Matafome

**Affiliations:** 1Coimbra Institute of Clinical and Biomedical Research (iCBR) and Institute of Physiology, Faculty of Medicine, University of Coimbra, 3000-548 Coimbra, Portugal; tamaehamonteiro@gmail.com (T.M.-A.); bia.caramelo@gmail.com (B.C.); daniarcha98@gmail.com (D.A.); andreia.amaro15@hotmail.com (A.A.); cat_barra@hotmail.com (C.B.); daniela.silva26@hotmail.com (D.S.); saraoliveira116@gmail.com (S.O.); rseica@fmed.uc.pt (R.S.); 2Center for Innovative Biomedicine and Biotechnology (CIBB), University of Coimbra, 3004-504 Coimbra, Portugal; 3Clinical Academic Center of Coimbra, 3000-548 Coimbra, Portugal; 4Research Group of Biotechnology and Bioprospecting Applied to Metabolism (GEBBAM), Federal University of Grande Dourados, Dourados 79825-070, MS, Brazil; 5Universitary Hospital Center of Coimbra, 3000-548 Coimbra, Portugal; 6Department of Complementary Sciences, Instituto Politécnico de Coimbra, Coimbra Health School (ESTeSC), 3046-854 Coimbra, Portugal

**Keywords:** natural and added sugars, fruit juices, hyperglycaemia, oxidative stress, glycation

## Abstract

Although fruit juices are a natural source of sugars, there is a controversy whether their sugar content has similar harmful effects as beverages’ added-sugars. We aimed to study the role of fruit juice sugars in inducing overweight, hyperglycaemia, glycation and oxidative stress in normal and diabetic animal models. In diabetic Goto-Kakizaki (GK) rats, we compared the effects of four different fruit juices (4-weeks) with sugary solutions having a similar sugar profile and concentration. In vitro, the sugary solutions were more susceptible to AGE formation than fruit juices, also causing higher postprandial glycaemia and lower erythrocytes’ antioxidant capacity in vivo (single intake). In GK rats, ad libitum fruit juice consumption (4-weeks) did not change body weight, glycaemia, oxidative stress nor glycation. Consumption of a matched volume of sugary solutions aggravated fasting glycaemia but had a moderate impact on caloric intake and oxidative stress/glycation markers in tissues of diabetic rats. Ad libitum availability of the same sugary solutions impaired energy balance regulation, leading to higher caloric intake than ad libitum fruit juices and controls, as well as weight gain, fasting hyperglycaemia, insulin intolerance and impaired oxidative stress/glycation markers in several tissues. We demonstrated the distinct role of sugars naturally present in fruit juices and added sugars in energy balance regulation, impairing oxidative stress, glycation and glucose metabolism in an animal model of type 2 diabetes.

## 1. Introduction

Increased consumption of westernized diets, highly processed foods and sugar-sweetened beverages is associated with an increased risk for obesity and associated pathologies throughout life. Such lifestyle changes are related to the health, economic and social burden of metabolic syndrome-associated diseases, such as type 2 diabetes, non-alcoholic fatty liver disease, cardiovascular diseases, among others. Dietary changes include higher consumption of processed foods rich in long-chain saturated fatty acids and sugars (soft drinks, snacks/desserts sweets, fast food, etc.), instead of fish, fruits and vegetables, which is apparently associated to weight gain [1,2]. Added sugars contribute to an higher energy density of the diet, leading to a positive energy balance and higher waist circumference, weight gain and development of metabolic disorders [3,4,5,6,7,8,9,10,11,12]. The World Health Organization (WHO) introduced a definition of free sugars (which includes sugars naturally present in fruit beverages) and established guidelines for free sugar intake in adults and children below 5–10% of the total daily energy [13]. According to the National Portuguese Food and Physical Activity Survey report, the national average consumption of simple sugars (mono and disaccharides) is 90 g (g)/day, contributing to an average of 19.8% for the total energy value [14].

Natural or intrinsic sugars are those naturally present in foods such as fruit and vegetables (fructose), honey and the sugars present in dairy products (galactose and (lactose). Besides sugars, these foods have several other compounds in their composition, which some of them regulate sugars absorption, cell uptake and metabolism [3]. The added sugars include mono and disaccharides that are added to foods during processing, preparation or at the table, with the objective of sweetening and increasing the food palatability and shelf life, improve texture, inhibit growth of microorganisms, give functional structures or give more accessibility [4,5,9,11]. Added sugars mainly include yellow sugar, corn sweetener, corn syrup, dextrose, fructose, glucose, high fructose corn syrup, lactose, maltose, malt sugar, molasses, raw, turbined sugar, trehalose, and sucrose [12,15]. They are mostly found in sugary drinks, pastry products, cookies, energy drinks, nectars, white bread and breakfast cereals [5,6,7].

Hyperglycaemia is related to the increased formation of advanced glycation end-products (AGEs), which are closely associated with the development and progression of diabetes and its complications [16,17,18,19,20]. In addition, AGEs may also be formed in sugar-rich foods or after exposure to high temperatures and low humidity during cooking [21]. AGEs are also involved in the development of insulin resistance–associated pathologies such as cardio- and cerebrovascular diseases, non-alcoholic steatohepatitis, and central nervous system disorders [22,23]. Vascular aging due to AGEs exposure, or vascular AGEing, is related to oxidative stress due to increased generation of reactive species of oxygen and nitrogen [22,24,25,26,27,28,29,30,31], endothelial dysfunction [28,29,30], and changes in the extracellular matrix [29] and in inflammatory factors [31].

The main objective of this study was to determine the distinct role of naturally present sugars in fruit juices and added sugars in impairing oxidative stress, glycation, glucose metabolism and energy balance in an animal model of type 2 diabetes. Our first specific objective was to determine the post-prandial glycaemia and total antioxidant capacity of erythrocytes immediately following the intake of different fruit juices and sugary solutions with a similar profile of glucose, fructose and sucrose. Our second objective was to compare the impact of chronic ad libitum fruit juice intake with the same profile, concentration, and quantity of added sugars on glucose metabolism and oxidative stress and glycation markers in erythrocytes, liver, adipose tissue, kidney, and heart of Goto-Kakizaki (GK) rats, a model of type 2 diabetes. The third objective was to disclose the role of both sugar sources in the regulation of energy balance, comparing ad libitum fruit juice intake with ad libitum intake of the same sugary solutions with matched sugar concentrations and profiles.

## 2. Materials and Methods

### 2.1. Chemicals and Antibodies

Salts and organic solvents used in this study were all purchased from Sigma-Aldrich/Merck Portugal (Oeiras, Portugal) and Fischer Scientific (Pittsburgh, PA, USA). Antibodies were used to target catalase, GLO-1 (Ab76110, Ab96032, Abcam, Cambridge, UK), CML (KH024, TransGenic Inc, Tokyo, Japan), MG-H1 (HM5017, Hycult Biotech, Uden, Netherlands) and argpyrimidine (AGE06B, Nordic-MUbio, Susteren, the Netherlands). Calnexin (AB0037, Sicgen, Cantanhede, Portugal) was used as loading control.

### 2.2. Fruit Juice and Sugars Samples

Fruit juice samples were kindly provided by SUMOL + COMPAL S.A. (Carnaxide, Portugal) Fruit juices were prepared from 100% fruit and no sugars were added during the process. In order to avoid sample-specific effects, four distinct samples were prepared: (1) red fruits (25% red grape, 16% raspberry, 16% apple, 11% strawberry, 11% plum, 9% pear, 6.5% banana, 5.5% blueberry); (2) orange (100% orange); (3) peach (50% peach, 20% apple, 15% mango, 5% apricot, 5% banana); and (4) pear (80% pear, 13% grape, 4% orange, 3% banana). Samples were produced at industrial scale. For each sample, the sugars profile was determined (Figure 1B) and a matched sugary solution was prepared daily for each sample, with the equivalent concentration of glucose, fructose and sucrose, and with similar pH to the respective juice sample. 

### 2.3. In Vitro Determination of AGEs Formation from Fruit Juices and Sugars’ Solutions

Samples of fruit juices were incubated at 4 °C or 37 °C during 30 days in the dark. Matched sugary solutions were incubated under the same conditions, in the presence or not of BSA. The concentration of BSA used was matched with the concentration of total protein in each of the juice samples. After the period of incubation, samples were stored at −80 °C for dot blot analysis using specific antibodies.

### 2.4. Determination of Post-Prandial Glycaemia and Total Antioxidant Capacity

The study was performed according to good practices of animal handling, with the approval of the Institutional Animal Care and Use Committee (ORBEA 13/2018) and the procedures were performed by licensed users of Federation of Laboratory Animal Science Associations—FELASA, conformed to the guidelines from the Directive 2010/63/EU of the European Parliament for the Protection of Animals Used for Science Purpose. 12-week-old Wistar rats from our breeding colony (Faculty of Medicine, University of Coimbra) were fed a specific volume (4 mL) of each fruit juice sample or the same volume of matched sugary solution (Figure 1B,C). Glycaemia was determined using a glucometer and reactive test stripes (Contour Next, Bayer Portugal, Lisboa, Portugal) before and 15, 30, 60, 90 and 120 min after gavage (*n* = 12/condition). In a different set of animals, the same protocol was followed, and blood samples were collected from the tail vein to Vacuette K3EDTA tubes (Greiner Bio-one, Kremsmünster, Austria) for evaluation of total antioxidant capacity. Blood samples were immediately centrifuged (2200× *g*, 4 °C, 15′) and the cellular fraction was diluted in the same volume of ultrapure H_2_O and submitted to repeated cycles of freeze/thawing. Supernatant was stored at −80 °C and later used for the Total Antioxidant Capacity Assay Kit (ab65329, Abcam) according to the manufacturer’s instructions.

### 2.5. Animal Maintenance and Treatment

12-week-old male Wistar rats were randomly divided in five groups (*n* = 6–8); control, red fruits juice (Wistar Red Fruits), orange juice (Wistar Orange), peach juice (Wistar Peach) and pear juice (Wistar Pear). Rats were maintained with free (ad libitum) access to the respective fruit juice (except the control, which received water) during 4 weeks (Figure 2A). Age-matched Goto-Kakizaki (GK) rats, a non-obese model of type 2 diabetes, were randomly divided in 13 groups (*n* = 6–8). Four groups were treated ad libitum with the same juices (GK Red Fruits, GK Orange, GK Peach, GK Pear). Four other groups were treated with a sugary solution equivalent in concentration, sugar profile and quantity the daily volume consumed by the respective groups treated with each fruit juice (GK RF_S, GK Orange_S, GK Peach_S, GK Pear_S). Another four groups were treated with sugary solutions with the same concentration and profile but with ad libitum access (GK Red Fruits_S_AL, GK Orange_S_AL, GK Peach_S_AL, GK Pear_S_AL). The last group (GK) was kept as control with free access to water and food. Animals were kept under standard conditions—2 animals per cage, with temperature at 22–24 °C, and 50–60% humidity, under standard light cycle (12 h light/12 h darkness). All animals were kept with ad libitum access to water and food (standard diet A03, Panlab, Barcelona, Spain) [32].

### 2.6. In Vivo Data and Sample Collection

Body weight, fasting glycaemia, water/juice/sugary solution and food intake were evaluated weekly. Before and after the treatment, animals were submitted to an intraperitoneal insulin tolerance test (ipITT) after 6 h of fasting. For the IIT, insulin was administered (i.p.) 0.1 U.kg^−1^ (Humulin, 1000 UI/mL Lilly, Lisboa, Portugal), followed by glycaemia measurement from the tail vein with a glucometer (Contour Next, Bayer) and test strips in time 0, 15, 30, 60 and 120 min. Response to insulin tolerance was expressed by area under the curve—AUC [33].

One day after the ipITT, 6-h fasted rats were anesthetized (i.p.) with 2:1 (*v*/*v*) 50 mg kg^−1^ ketamine (100 mg/mL)/2.5% chlorpromazine (5 mg mL^−1^) and samples of blood was collected by cardiac puncture followed by cervical dislocation. Epididimal adipose tissue (EAT), liver, kidney and heart were collected for further analysis. After centrifugation of blood samples, cell fraction was diluted in the same volume of H_2_O, submitted to freeze/thawing cycles and the supernatant was used for the determination of the total antioxidant capacity (ab65329, Abcam).

### 2.7. Western Blot and Dot Blot

Tissues were disrupted in lysis buffer (0.25 M Tris-HCl, 125 mM NaCl, 1% Triton-X-100, 0.5% SDS, 1 mM EDTA, 1 mM EGTA, 20 mM NaF, 2 mM Na_3_VO_4_, 10 mM β-glycerophosphate, 2.5 mM sodium pyrophosphate, 10 mM PMSF, 40 µL of protease inhibitor), centrifuged (14,000 rpm, 20 min, 4 °C) and denaturated with Laemmli buffer (62.5 mM Tris-HCl, 10% glycerol, 2% SDS, 5% β-mercaptoethanol, 0.01% bromophenol blue). Protein was quantified through the BCA Protein Assay Kit [34]. Samples were loaded in SDS-PAGE and electroblotted onto PVDF membrane (Advansta, San Jose, CA USA). For dot blot, 1 μL of denaturated samples were directly leaded to nitrocellulose membranes. All membranes were blocked with TBS-T 0.01% and BSA 5%, then incubated with the primary and respective secondary antibodies anti-mouse (GE Healthcare, Chicago, IL, USA), anti-rabbit and anti-goat (Bio-Rad Portugal, Lisboa, Portugal). Calnexin was used as loading control. Immunoblots were detected with ECL substrate and the Versadoc system (Bio-Rad).

### 2.8. Dihydroethidium (DHE) Staining

The evaluation of kidney oxidative stress was performed through the detection of dihydroethidium (DHE) in cryosections (5 mm). DAPI was used to stain nucleus and images were obtained with a fluorescence microscope (Zeiss Axio Observer Z1) equipped with an incorporated camera (Zeiss, Jena, Germany), detected with 587 nm of excitation and 610 nm of emission for DHE, and 353 nm of excitation and 465 nm of emission for DAPI. The same settings were kept constant for all analysis.

### 2.9. Statistical Analysis

Data were expressed as the mean ± standard error of the mean (SEM) and compared by analysis of variance (ANOVA) followed by Tukey post-hoc test. *p* < 0.05 was considered significant. Statistical tests were performed with GraphPad Prism 5.0 (GraphPad, San Diego, CA, USA) and IBM SPSS Statistics Software (IBM, Armonk NY, USA).

## 3. Results

### 3.1. Sugars Naturally Present in Fruit Juices Are Less Prone to AGEs Formation Than Added Sugars in Sugary Solutions and Cause a Lower Increase of Postprandial Glycaemia

The AGEs MG-H1, argpyrimidine and CML were indetectable by dot blots in all fruit juices samples incubated for 30 days at 4 °C or 37 °C (Figure 1A). On the other hand, the incubation of each sugary solution with the respective amount of BSA (the same protein content of each juice sample) resulted in immune staining of all AGEs. The intensity of the signal for each AGE depended on the type of sugar profile and temperature of incubation (Figure 1A). MG-H1 was more detected in samples incubated at 4 °C and Argpyrimidine was more detected in samples incubated at 37 °C. Such observations are consistent with the fact the MG-H1 is a more reversible product of methylglyoxal reaction with proteins, while Argpyrimidine is a more final and irreversible product of such reaction. No AGE formation was detectable when sugars were incubated without protein (Figure 1A).

The intake of a specific volume of each juice sample produced a rapid increase of glycaemia in relation to the sham rats (same volume of water). The less evident result was observed for pear, which has a lower concentration of glucose (Figure 1D–G). Nevertheless, such increase was higher for every corresponding sugary solution. Such higher increase of post-prandial glycaemia with the sugary solutions was associated with a lower total antioxidant capacity of erythrocytes at the same time-points after ingestion (Figure 1H–K).

### 3.2. Added Sugars Do Not Cause Satiety and Lead to Higher Body Weight, Caloric Intake and Impaired Glycemic Profile When Available Ad Libitum

In both normal and diabetic rats, the ad libitum consumption of fruit juices did not results in a significant body weight gain in relation to their respective controls. As well, the consumption of a matched sugary solution by diabetic rats did not change weight gain. Weight gain was only increased in the groups of diabetic rats treated with sugary solutions ad libitum, achieving statistical significance for pear juice solution (Figure 2B–I).

This may be attributable to a poorer control of energy balance. Diabetic rats treated with fruit juices (ad libitum) drank significantly more liquid volume than controls (water) (Figure 2N–Q), but significantly reduced their food intake (Figure 2J–M), resulting in a moderate non-significant (significant only for the peach juice sample, *p* < 0.05 vs GK) increase of caloric intake (Figure 2R–U). Rats treated with the matched sugary solutions, besides drinking all the volume supplied, did not reduce their food intake (Figure 2J–M), leading to significantly higher caloric intake for almost all the samples (Figure 2R–U). When the same sugary solutions were supplied ad libitum, the water intake was further increased (Figure 2N–Q), although the food consumption was not reduced (Figure 2J–M). This resulted in a significantly higher caloric intake than GK rats maintained with water (Figure 2R–U), which was associated with increased body weight gain throughout the treatment (S_AL).

GK rats are a spontaneous model of type 2 diabetes, showing fasting hyperglycaemia and decreased insulin tolerance (higher AUC during the ITT) (Figure 3). The consumption of fruit juices did not cause significant changes in fasting (6 h) glycaemia throughout the experimental period in relation to the initial value (Figure 3A–D), nor at the end in relation to GK rats maintained with water (Figure 3E–H). In the case of orange juice, fasting glycaemia was in fact reduced in relation to GK rats at the end of the treatment (Figure 3F). Fruit juices also did not change insulin tolerance at the end of the experimental period (Figure 3I–L). The consumption of a matched (profile and volume) sugary solution slightly increased fasting glycaemia and insulin resistance at the end of the treatment, with a more significant effect being observed for red fruits sugars, possibly because they are richer in glucose (Figure 3E–H). The ad libitum consumption of the same sugary solutions produced an increase of fasting glycaemia throughout the experimental period in relation to the initial value and to the GK maintained with water (Figure 3A–D), as well as at the end of the treatment in relation to GK rats maintained with water and treated with fruit juices (Figure 3E–H). Such consumption also resulted in a significant deterioration of insulin tolerance, with higher AUC during the ITT (Figure 3I–L).

### 3.3. Consumption of Added Sugars Further Impair Glycation and Oxidative Stress Markers in Liver, Adipose Tissue, Heart and Kidney

No significant differences were observed for total antioxidant capacity of erythrocytes in any group (Figure 4A–D). In erythrocytes, the levels of CML were increased in the red fruits juice group, but only in diabetic rats (*p* < 0.05 vs GK). Nevertheless, CML levels were further increased by the consumption of the sugary solution, either in the same amount of fruit juices or ad libitum (*p* < 0.01 vs GK) (Figure 4E). Such increased levels were also observed after ad libitum consumption of the sugary solution similar to pear juice (Figure 4H). Increased erythrocytes argpyrimidine was observed in diabetic rats after consumption of orange, peach and pear juice and was maintained or further increased in rats maintained with the sugary solutions either in the same amount of fruit juices or ad libitum (Figure 4I–L).

The ad libitum consumption of sugary solutions caused an increase of liver weight in diabetic rats (Figure 5A–D). Diabetic rats showed a downregulation of catalase, which, taking all the fruit juices together, was not significantly changed by the consumption of fruit juices or sugary solutions (Figure 5E–H). Besides, there were no observed changes in GLO-1 levels, nor argpyrimidine and CML in the dot blot staining (Figure 5I).

Similar results were observed in visceral adipose tissue, showing no significant differences between groups in argpyrimidine and CML (Figure 6M). GK rats had lower visceral adipose tissue weight (Figure 6A–D), catalase expression (Figure 6E–H) and tendentially lower GLO-1 expression (Figure 6I–L). Fruit juice consumption did not change adipose tissue weight or GLO-1 expression and induced a decrease of catalase expression in diabetic rats. 

On the other hand, consumption of sugary solutions either in the same amount of fruit juices or ad libitum resulted in a reduction of adipose tissue weight (Figure 6A–D), a marker of adipose tissue dysfunction, and a further reduction of catalase (Figure 6E–H) and GLO-1 levels (Figure 6I–L) in relation to GK rats maintained with water. Thus, consumption of fruit juices did not change adipose tissue and liver weight, nor changed the lower expression of antioxidant and antiglycant enzymes. Although no differences were observed for AGEs levels in both tissues, the caloric contribution of sugary solutions conduced to liver hypertrophy in diabetic rats, instead of adipose tissue expansion, which was associated with a further downregulation of both antioxidant enzymes.

Diabetic rats had lower heart weight, which was not changed by fruit juices consumption. Instead, the consumption of sugary solutions resulted in an increase of heart weight, being more evident in rats maintained with ad libitum access (Figure 7A–D). In heart, the consumption of sugary solutions led to a compensatory increase of catalase expression, especially when available ad libitum, achieving statistical significance in sugars similar to orange profile (Figure 7E–H). Such increase in catalase levels in rats maintained with *ad libitum* access to sugars may be attributable to the stress caused by the excessive sugar intake. No differences were observed for GLO-1 levels (Figure 7I). The levels of argpyrimidine were not changed by the consumption of fruit juices, but significantly increased in response to sugary solutions, namely with red fruits and pear sugar profile (Figure 7J–M). Similarly, the levels of CML were also increased after consumption of the sugary solutions, namely with red fruits, orange and pear sugars profile, while no major differences were observed after juice consumption (Figure 7N–Q). The heart is particularly susceptible to lipid peroxidation and 8-isoprostane was used as a tissue biomarker of such mechanisms. GK rats have tendentially higher 8-isoprostane levels in the heart, which were in fact reduced by the consumption of fruit juices, especially the peach sample (Figure 7R–U). The rats treated with the sugary solutions revealed 8-isoprostane levels similar to diabetic rats maintained with water. 

In kidney, the consumption of fruit juices did not change the weight of the organ in diabetic rats, which was significantly higher in rats maintained with the sugary solutions. Such increase was observed in rats with matched volume of solution but was especially observed in those with ad libitum access (Figure 8A–D). No significant changes were observed in catalase, GLO-1 and CML levels (Figure 8E), while an increase of argpyrimidine levels was observed in rats maintained with the sugary solution similar to pear juice, the one with higher fructose content (Figure 8F–I).

The existence of oxidative stress particularly in the glomerulus, was assessed through the staining with DHE, a probe for superoxide anion. No changes in DHE reactivity were observed in Wistar rats maintained with fruit juices, in relation to the controls (Figure 9A–C). The diabetic rats maintained with fruit juices revealed a increase of glomerular DHE staining the was only significant for ref fruit sample (Figure 9E,H–K). On the other hand, diabetic rats maintained with the matched volume of the sugary solutions and especially those maintained with ad libitum access to the same solutions revealled a significant increase of DHE staining, being observed in all the samples tested (Figure 9F–K).

## 4. Discussion

In this study, we have demonstrated the distinct role of sugars naturally present in fruit juices and added sugars in impairing oxidative stress, glycation, glucose metabolism and energy balance in an animal model of type 2 diabetes. We have compared sugary solutions with a similar profile of glucose, fructose and sucrose to four different samples of fruit juices. We have shown that sugary solutions induce a higher post-prandial glycaemia and lower total antioxidant capacity of erythrocytes when comparing to the corresponding fruit juices samples. We have also shown that the chronic intake of the same sugary solutions by type 2 diabetic rats, especially when available ad libitum, leads to an imbalance of energy intake regulation, causing increased body weight gain and hyperglycaemia. Such alterations are associated with increased markers of glycation and oxidative stress in several organs. Altogether, heart and kidney appear to be more susceptible to oxidative stress and glycation after an ad libitum consumption of added sugars, while fruit juices were not associated with such alterations.

Consumption of fast-food has been considered in many observational studies as a risk factor for being overweight or obese in both developed and developing regions, even after controlling for energy intake [35]. Higher intake of sugar sweetened drinks was associated with poorer dietary choices and correlated with higher BMI and waist circumference in children [36]. In fact, a recent study by Fox and colleagues has shown increased children’s sugar intake from added sources in school breakfasts and lunches [37]. About 63% of children were shown to surpass the limit defined by the Dietary Guidelines for Americans for 24 h intake of added sugars [37]. In adults, increased daily dietary glycemic index was positively associated with higher BMI, with little effect of the percentage of calories from total carbohydrates or total carbohydrate consumption [38]. Increased energy density of foods and macronutrients consumption are usually considered the main determinants for adiposity and metabolic syndrome. However, the source of nutrients are apparently an important factor to take into account, mainly because of other compounds/trace elements that are often present in many foods [39,40]. Such compounds may influence nutrient digestion, absorption, metabolism or biochemical effect, and may be absent in foods or beverages with high amounts of added nutrients (sugars or fat) or may be lost during their industrial processing. The consumption of processed foods with higher sugar and fat content has been suggested as a risk factor for the development of metabolic syndrome components [39]. On the other hand, the consumption of unprocessed natural foods has been suggested as a protective dietary strategy to improve metabolic syndrome components and biochemical markers of disease [39,40].

Consumption of sugar-sweetened beverages (SSB) and fructose-rich fruit juices have been associated with markers of cardiometabolic disease, such as triglycerides (sugars are used for de novo lipogenesis) and HbA1c, as well as with increased risk of arthritis and asthma in pediatric populations, possibly due to increased fructose-induced AGEs formation, RAGE activation and inflammation [41,42,43,44,45]. A recent meta-analysis concluded that increased energy density of meals and food source (processed foods) are key determinants of fructose impact on glycemic control and insulinemia [46]. Data supported that natural sources of fructose-containing sugars or their matched substitution for other macronutrients do not have a negative impact on HbA1c [46]. On the other hand, the addition of the same sugars has harmful effects. The authors disclosed that natural sources like fruit or fruit juices have protective effects on glycemic control, while foods like sweetened milk with added sugars have negative effects [46]. In the present study, fruit juices were supplied ad libitum and no restraint on their consumption was imposed during the treatment period. Remarkably, we show here that ad libitum consumption of fruit juices did not have a negative impact on body weight, glycaemia, glycation and oxidative stress, even in diabetic rats. Such effects were not distinct from those observed in normal rats with ad libitum acess to the same fruit juices. When compared to control Wistar rats, the consumption of fruit juices by Wistar rats did not change any of the parameters studied. Control Wistar rats were also used as normal model to compare the effects of fruit juices and sugary solutions in GK rats. Although 14-weeks-old GK rats are not significantly different from Wistar rats, the consumption of sugary solutions, especially when ad libitum, further increased such differences.

One of the mechanisms for the protective effects of natural sugar sources like fruit juices is the presence of other compounds with the capacity to prevent glycation and oxidative stress. For instance, previous studies have shown that fructose-induced glycation of albumin in vitro was 98% inhibited by pomegranate juice, but less by other fruit juices. Orange, grape or cranberry juice inhibited glycation by 20%, while apple juice didn’t confer any protection [47]. Accordingly, citrus juice has been suggested to have beneficial effects on several cardiometabolic markers on patients with type 2 diabetes [48]. On the contrary, beverages or sweetened fruit juices were shown to have higher amounts of AGEs than natural fruit juices [49]. Fruit juices are a natural source of flavonoids and anthocyanins. They have been suggested by several authors as useful in regulating glycaemia and as good supplements or treatment for type 2 diabetes [50,51,52]. Authors have suggested that such effects may, not only rely in the direct antioxidant properties of the molecules, but also in their ability to modulate mitochondrial function or gut microbiota diversity [53,54].

Another important factor for the protective effects of fruit juices when compared to beverages or sweetened fruit juices is the modulation of satiety mechanisms. Although physical fruit processing for juice preparation has been shown to accelerate gastric emptying and to potentially reduce satiety, the presence of fiber in natural fruit juices was shown to reduce postprandial hyperglycaemia, while increasing the feelings of satiety and fullness [55,56]. Importantly, we here show that fruit juices, besides being a natural source of sugars, have an impact on satiety mechanisms, given that their ad libitum consumption is associated with lower food intake, especially in diabetic rats. Such mechanisms are absent when consuming the corresponding sugary solutions, given that rats fed with matched solutions, not only do not decrease food intake, but actually increase it in most of the samples tested. Remarkably, ad libitum availability of such solutions leads to their significantly higher consumption, also not leading to lower food intake. Altogether, this results in a energy imbalance towards energy intake that leads to weight gain, hyperglycaemia, oxidative stress and glycation. Spetter et al., have shown that fruit juices consumption leads to an inhibition of striatum activation. More, they have shown that activation of anterior cingulate area in the central nervous system predicts subsequent reduction of fruit juices consumption, suggesting a food-specific satiety [57]. Indeed, the role of AGEs and oxidative stress in the hypothalamus must also be considered as a possible mechanisms in the dysregulation of energy imbalance, which needs further investigation. Although little is known regarding the activation of brain areas in response to different sugars composition of foods, it is possible that added sugars, out of a natural matrix with other substances that potentially retard their intestinal absorption and protect from their harmful effects, may cause a distinct impact on such brain areas. Importantly, the pleasantness induced by different sugar content in fruit juices may affect the activation of such areas and can also be modified along the time in order to reduce sugar intake. It was demonstrated that reduction of added sugar to orange juices is well tolerated by the consumers without affecting acceptance [58].

In summary, fruit juices are a natural source of sugars. However, they are ingested in a complex mixture of other nutrients, fibers and other known and unknown compounds that may delay their digestion and absorption, preventing postprandial hyperglycaemia and adverse metabolic effects. Our results show that added sugars have a distinct impact on glycemic control. Matched sugars to four different fruit juices samples do not cause a significative difference on glycaemia, glycation and oxidative stress, even in a diabetic model. Even so, our results show that added sugars have a completely different effect in the modulation of satiety and regulation of energy balance, leading to a poorer glycemic profile and increased levels of glycation and oxidative stress markers, particularly in tissues like the heart and the kidney.

A limitation of our study is the relative short time of juice/sugary solution administration. Future studies could perform administrations for longer periods, in order to undertand the long-term effects of naturally present and added sugars. Longer administrations in more aged diabetic models could reveal more harmful effects of sugars in later stages of disease. Another question that should be answered in the future is the consequences for insulin signalling in liver and epripheral tissues, as well as for insulin secretion and beta-cell viability.

In conclusion, our results reinforce the evidence supporting a noxious role for added sugars in foods, especially when in aqueous solutions, and a harmless effect of fruit sugars without added sugar when consumed moderately, even in diabetic models. We believe that such evidence will help to create the awareness for the need of better food policies and nutritional advices for the general population.

## Figures and Tables

**Figure 1 nutrients-13-02956-f001:**
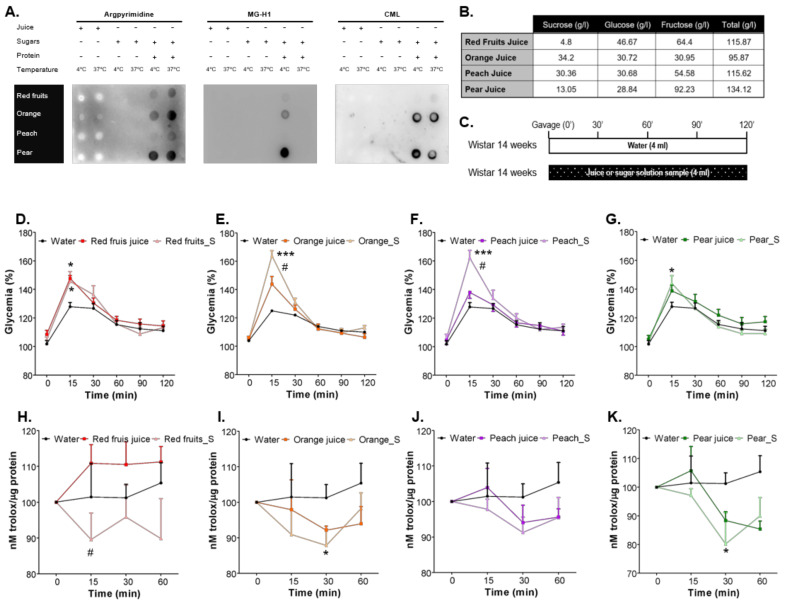
The AGEs CML, MG-H1 and Argpyrimidine were detected by dot blot (**A**) in fruit juices and sugary solutions with the same sugars profile (**B**) in the presence or absence of BSA, at 4 °C or 37 °C. In other to determine the postprandial glycaemia, an OGTT was performed (**C**) and glycaemia and total antioxidant capacity of erythrocytes was determined after intake of 4 mL of red fruits (**D**,**H**), orange (**E**,**I**), peach (**F**,**J**) and pear (**G**,**K**) fruit juices or a sugary solution with the equivalent sugars profile. * different from Water; # different from the fruit juice. 1 symbol, *p* < 0.05; 3 symbols, *p* < 0.001.

**Figure 2 nutrients-13-02956-f002:**
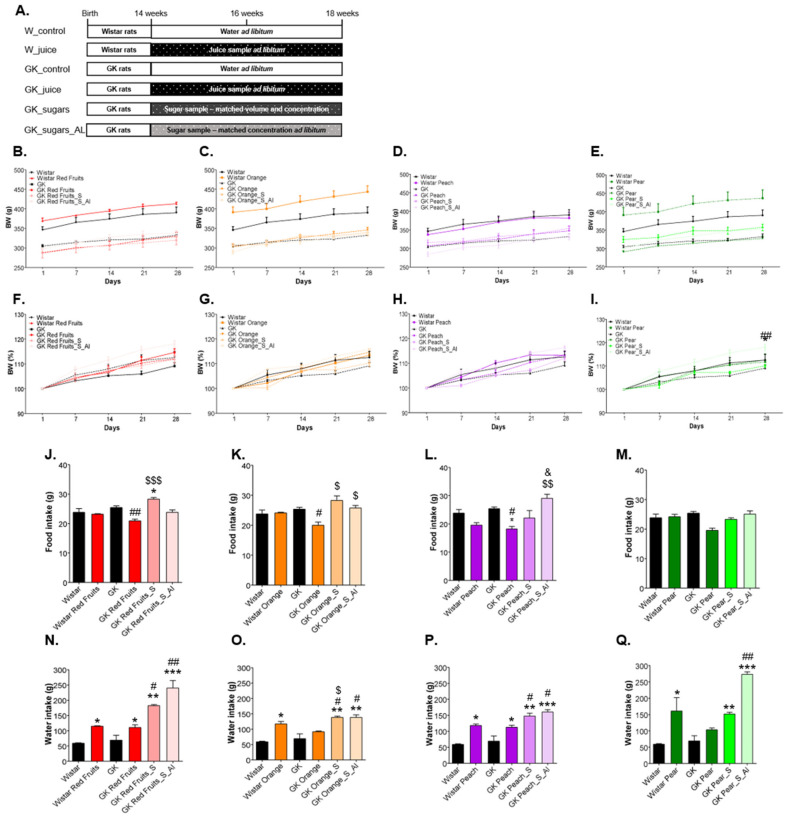
Normal and diabetic rats were treated for 4 weeks with fruit juices ad libitum. Diabetic rats were also treated during the same period with a matched sugary solution in the same volume (GK_S) or ad libitum (GK_S_AL) (**A**). Body weight was monitored during the experimental period (**B**–**E**) and weight gain calculated at each time-point (**F**–**I**). Food intake (**J**–**M**) and consumption of water/juice/sugary solution (**N**–**Q**) were monitored throughout all the experimental period. The total caloric intake/day was calculated (**R**–**U**). * different from Wistar; # different from GK; $ different from GK_Juice. & Different from GK_Juice_S. 1 symbol, *p* < 0.05; 2 symbols, *p* < 0.01; 3 symbols, *p* < 0.001.

**Figure 3 nutrients-13-02956-f003:**
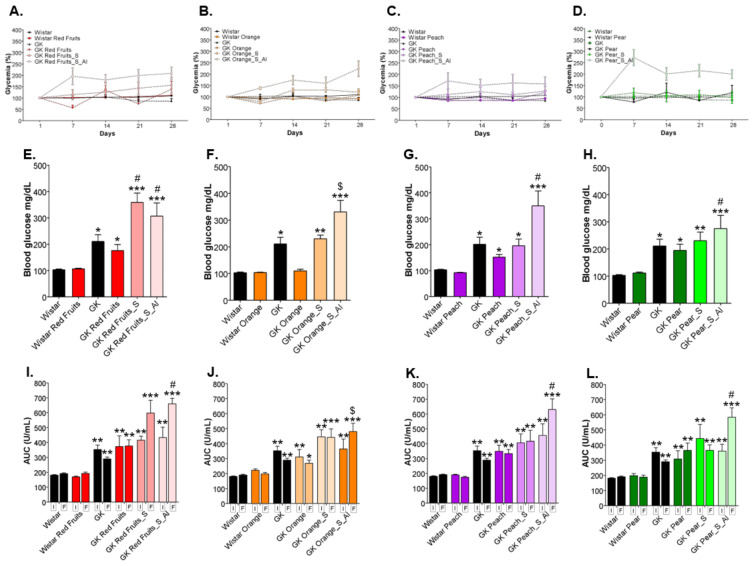
Fasting (6 h) glycaemia was evaluated weekly and calculated the percentage of the initial value (**A**–**D**). Fasting glycaemia at the end of the treatment is shown in (**E**–**H**). Before and after the experimental period an i.p. insulin tolerance test was performed and the area under the curve was calculated (**I**–**L**). * different from Wistar; # different from GK; $ different from GK_Juice. 1 symbol, *p* < 0.05; 2 symbols, *p* < 0.01; 3 symbols, *p* < 0.001.

**Figure 4 nutrients-13-02956-f004:**
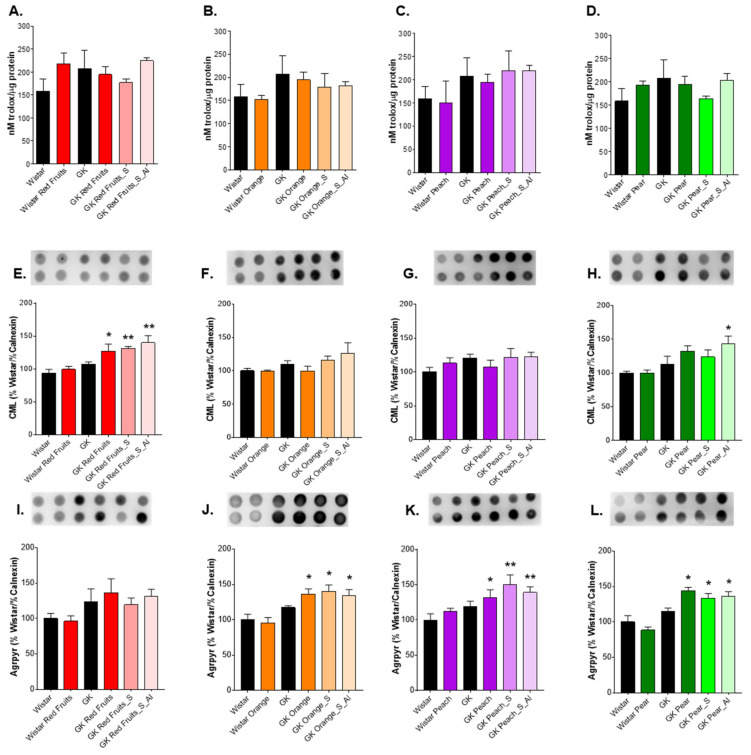
In isolated erythrocytes, total antioxidant capacity was determined (**A**–**D**) and CML (**E**–**H**) and argpyrimidine (**I**–**L**) were detected by dot blot. * different from Wistar. 1 symbol, *p* < 0.05; 2 symbols, *p* < 0.01.

**Figure 5 nutrients-13-02956-f005:**
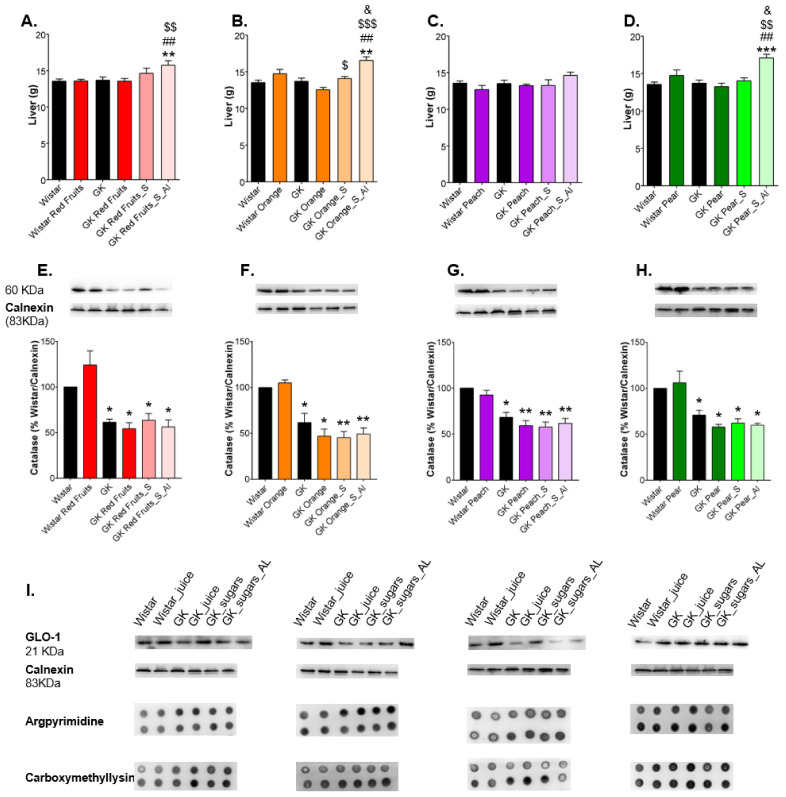
Liver weight was recorded (**A**–**D**) and the levels of catalase were determined by Western Blot (**E**–**H**). Representative western blot membranes of GLO-1 and dot blot membranes of CML and argpyrimidine in the liver are shown in (**I**). * different from Wistar; # different from GK; $ different from GK_Juice; & different from GK_S. 1 symbol, *p* < 0.05; 2 symbols, *p* < 0.01; 3 symbols, *p* < 0.001.

**Figure 6 nutrients-13-02956-f006:**
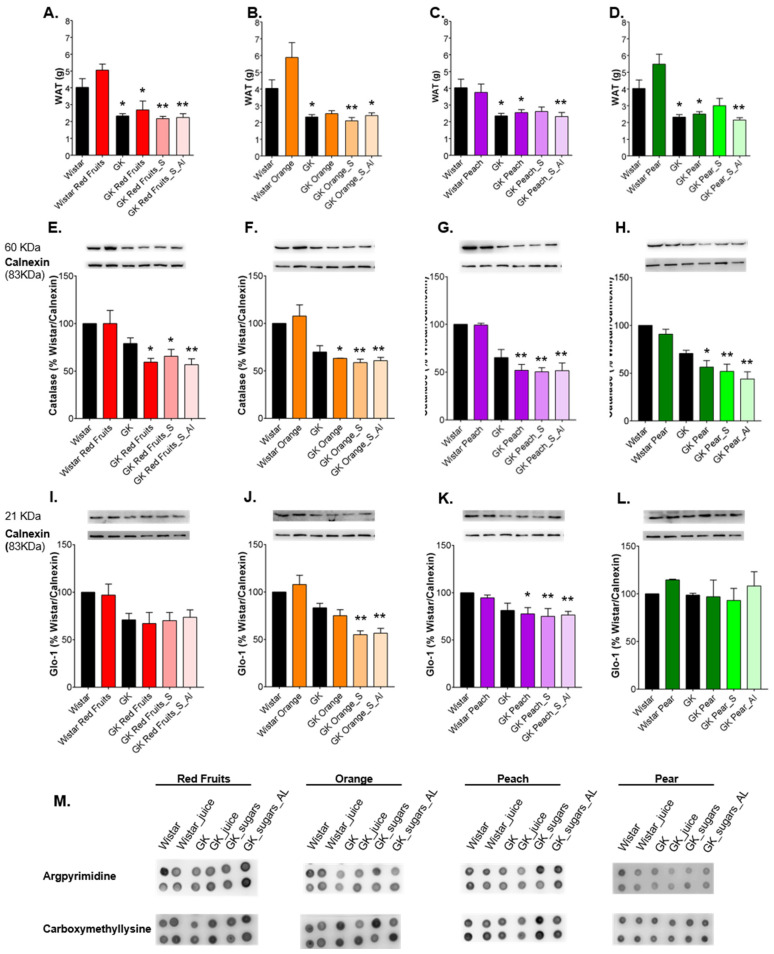
Epididymal adipose tissue weight was recorded (**A**–**D**) and the levels of catalase (**E**–**H**) and GLO-1 (**I**–**L**) in the tissue were determined by Western Blot. Representative dot blot membranes of CML and argpyrimidine in the adipose tissue are shown in (**M**). * different from Wistar. 1 symbol, *p* < 0.05; 2 symbols, *p* < 0.01.

**Figure 7 nutrients-13-02956-f007:**
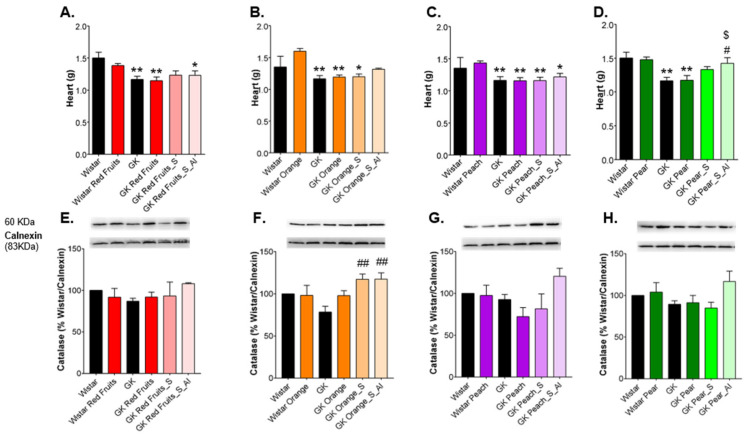
Heart weight was recorded (**A**–**D**) and the levels of catalase were determined by Western blot (**E**–**H**). Representative western blot membranes of GLO-1 in the heart are shown in (**I**). Heart CML (**J**–**M**) and argpyrimidine (**N**–**Q**) were determined by dot blot. 8-Isoprostane levels were determined as a marker of lipid peroxidation (**R**–**U**). * different from Wistar; # different from GK; $ different from GK_Juice. 1 symbol, *p* < 0.05; 2 symbols, *p* < 0.01.

**Figure 8 nutrients-13-02956-f008:**
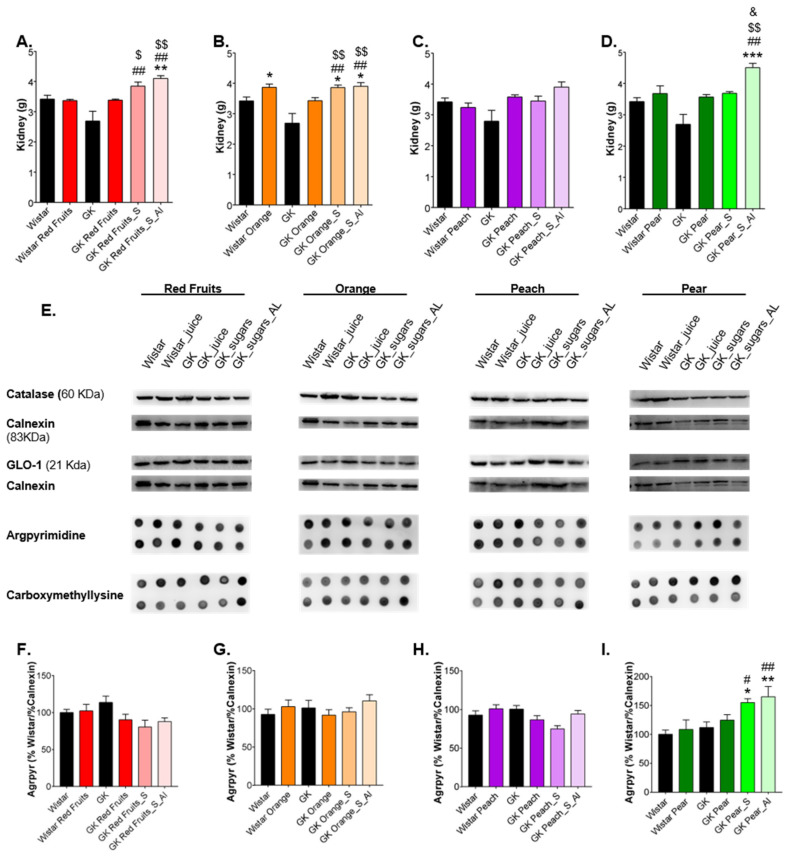
Kidney weight was recorded (**A**–**D**). Representative Western blot membranes of catalase and GLO-1 and dot blot membranes of CML in the kidney are shown in (**E**). Kidney argpyrimidine was determined by dot blot (**F**–**I**). * different from Wistar; # different from GK; $ different from GK_Juice; & different from GK_S. 1 symbol, *p* < 0.05; 2 symbols, *p* < 0.01.

**Figure 9 nutrients-13-02956-f009:**
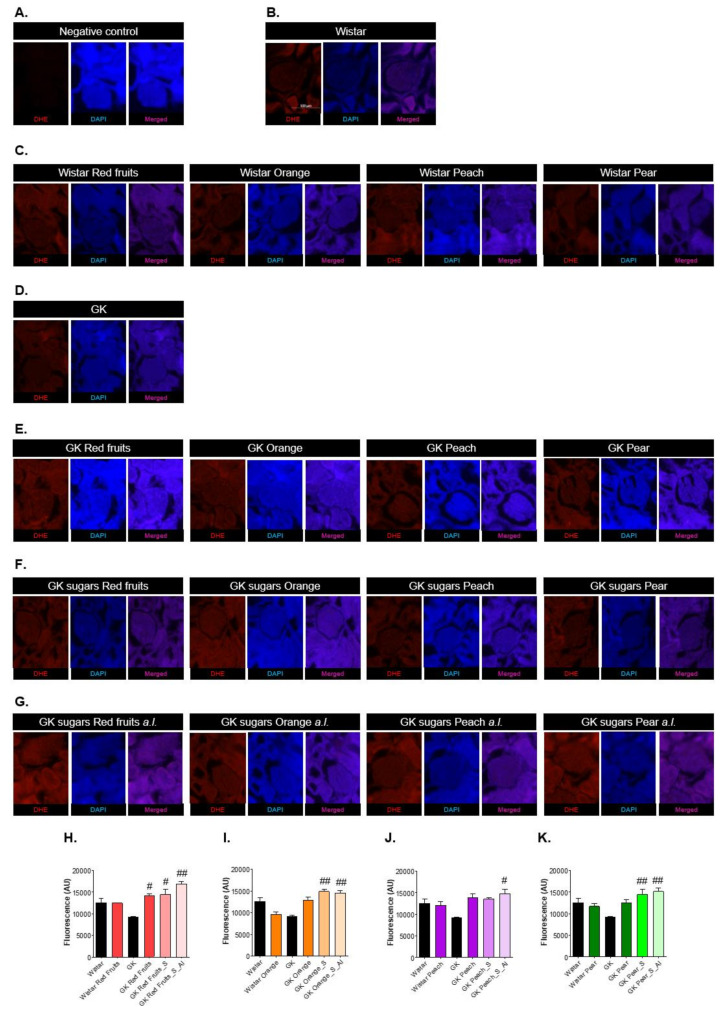
Detection of the dihydroethidium (DHE) probe for superoxide anion determination. (**A**) shows the negative control (no probe). Representative images of the glomeruli are shown for control Wistar rats (**B**), Wistar rats maintained with ad libitum access to fruit juices (**C**), control GK rats (**D**), GK rats maintained with ad libitum access to fruit juices (**E**), GK rats treated with respective sugary solutions matched in sugar profile, concentration and quantity (**F**), and GK rats maintained with ad libitum access to the same sugary solutions (**G**). (**H**–**K**) show the quantification of glomerular DHE staining for the different experimental conditions of each sample tested. # different from GK. 1 symbol, *p* < 0.05; 2 symbols, *p* < 0.01.

## Data Availability

The datasets generated during and/or analyzed during the current study are available from the corresponding author upon reasonable request.

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
