# Peer review of "Distinct Impact of Natural Sugars from Fruit Juices and Added Sugars on Caloric Intake, Body Weight, Glycaemia, Oxidative Stress and Glycation in Diabetic Rats"

_nutrients, 2021, doi:10.3390/nu13092956_

Round 1

Reviewer 1 Report

In this study, Monteiro-Alfredo and co-workers highlight the different impact of natural vs added sugar on glycaemic control, glycation, oxidative stress and satiety. Authors, concluded that when compared to added sugars, natural sugars present in fruit juices exert beneficial effects on the aforementioned parameters without exerting detrimental metabolic effects. The study is of particular interest as it focus on a really important topic in relation to public health nutrition and the exponential increase in obesity and associated cardio-metabolic disorders. Nevertheless, I have some concerns relative to this study. In particular, the rational of using two separate rat models is not defined. It is not clear why wild-type rats were not fed the sugary drink (added sugar)? Was food intake normalised for each group? This is of particular importance considering that sugary drinks and fruit juices may impact upon total food intake. It looks like that the experiments involving GK rats were “complete” including all three different interventions: water (control), fruit juices and sugary solutions. However, the same is not true for Wistar rats. Furthermore the quality of the writing needs to be extensively revised, to make the text clearer and easier to read. Additional concerns are reported below:

Line 36: it should be: “Western diet, highly processed foods…”.

Line 41: “fats” should be changed to “long-chain saturated fatty acids”.

Line 55: Reference number 3 does not seem to support the statement made by Authors.

Line 64-64: It should be noted that AGEs are also formed during food processing.

Line 126: What do Authors mean by: “submitted to repeated cycles of freeze/thawing”?

The experiment for the quantification of AGEs is not well balanced. While for the fruit juice the formation of AGEs was only dependent on endogenous protein content, in the sugary drinks there was an addition of BSA. It would be important to match the amount of BSA in the in the juices and sugary drinks to confirm that the increased formation of AGEs in the sugary drinks is not dependent on BSA.

Line 227: This statement only applies to GK rats as Wistar rats were not given the sugary solution.

Figures are not easy to follow as the graphical quality, at least with regard to the text provided to the reviewer, is very low. Es. Figure 2 B-I. This is particularly evident for x-y graphs.

What is the rationale of: Comparing Wistar and GK rats?

Line 252: What do Authors mean by “in relation to relation to GK rats”? it seems like Authors are already talking about GK rats. Please clarify.

Figure 9: Was the staining quantified?

Line 395: Please change “metabolization” to “metabolism”.

Line 405: It would also be important to refer to de novo lipogenesis.

Line 406-407: “energy density and food source” this requires clarification, because as it stands it is not clear.

Line 408: “they matched” it is not clear what this refers to.

In terms of the regulation of energy intake and energy balance, another important aspect to consider is the role of the hypothalamus and the putative ability of AGEs to promote hypothalamic inflammation and dysfunction.

Author Response

In this study, Monteiro-Alfredo and co-workers highlight the different impact of natural vs added sugar on glycaemic control, glycation, oxidative stress and satiety. Authors, concluded that when compared to added sugars, natural sugars present in fruit juices exert beneficial effects on the aforementioned parameters without exerting detrimental metabolic effects. The study is of particular interest as it focus on a really important topic in relation to public health nutrition and the exponential increase in obesity and associated cardio-metabolic disorders.

We are grateful to the reviewer for the careful evaluation of our manuscript, as well as for the constructive comments that we used to improve the manuscript quality. Please find below a point-by-point response to all the comments, questions and suggestions raised.

Nevertheless, I have some concerns relative to this study. In particular, the rational of using two separate rat models is not defined. It is not clear why wild-type rats were not fed the sugary drink (added sugar)? It looks like that the experiments involving GK rats were “complete” including all three different interventions: water (control), fruit juices and sugary solutions. However, the same is not true for Wistar rats.

Thank you for the considerations. We have used two separate models because the objective of the study was to compare fruit juices with sugary solutions in a diabetic model. Wistar rats were used only as controls. In fact, we have also administered fruit juices to normal rats because our initial hypothesis was that fruit juices would have some harmful effects in diabetic rats and Wistar rats treated with fruit juices would be the control groups for that. In fact, the effects of the fruit juices in diabetic rats were scarce but we opted by maintaining the Wistar-fruit juices groups in the manuscript. However, sugary solutions were only administered to diabetic rats because it was the main objective of the study. Please notice that studying all samples in Wistar rats would lead to eight more experimental groups, which would create a very confuse manuscript and decrease the focus of the study.

Was food intake normalised for each group? This is of particular importance considering that sugary drinks and fruit juices may impact upon total food intake.

No, we have not normalised food intake, because one of the objectives was to study the impact of fruit juices and sugary solutions on food intake and energy balance. In fact, that is one of the major conclusions of the study

The quality of the writing needs to be extensively revised, to make the text clearer and easier to read. Additional concerns are reported below:

The manuscript was carefully revised.

Line 36: it should be: “Western diet, highly processed foods…”.

Corrected

Line 41: “fats” should be changed to “long-chain saturated fatty acids”.

Corrected

Line 55: Reference number 3 does not seem to support the statement made by Authors.

Corrected

Line 64-64: It should be noted that AGEs are also formed during food processing.

The information was included

Line 126: What do Authors mean by: “submitted to repeated cycles of freeze/thawing”?

Freeze/thawing was used to lysate the cells.

The experiment for the quantification of AGEs is not well balanced. While for the fruit juice the formation of AGEs was only dependent on endogenous protein content, in the sugary drinks there was an addition of BSA. It would be important to match the amount of BSA in the juices and sugary drinks to confirm that the increased formation of AGEs in the sugary drinks is not dependent on BSA.

The BSA used was the same present in the fruit juice of each sample to match the total protein content. Such information was present in the first version of the manuscript “The concentration of BSA used was matched with the concentration of total protein in each of the juice samples”, in section 2.3

Line 227: This statement only applies to GK rats as Wistar rats were not given the sugary solution.

Corrected

Figures are not easy to follow as the graphical quality, at least with regard to the text provided to the reviewer, is very low. Es. Figure 2 B-I. This is particularly evident for x-y graphs.

The font size was increased in order to improve readability

What is the rationale of: Comparing Wistar and GK rats?

Wistar rats were used as the control situation for the diabetic model.

Line 252: What do Authors mean by “in relation to relation to GK rats”? it seems like Authors are already talking about GK rats. Please clarify.

It refers to the control GK rats. The information was included

Figure 9: Was the staining quantified?

No, we opted by not quantifying the staining because we used order quantitative or semi-quantitative markers, namely AGEs and antioxidant enzymes. Quantifying fluorescence may produce biased results so we opted for only showing the images obtained.

Line 395: Please change “metabolization” to “metabolism”.

Corrected

Line 405: It would also be important to refer to de novo lipogenesis.

The information was included

Line 406-407: “energy density and food source” this requires clarification, because as it stands it is not clear.

The sentence was clarified

Line 408: “they matched” it is not clear what this refers to.

Should be “their matched”. It was corrected.

In terms of the regulation of energy intake and energy balance, another important aspect to consider is the role of the hypothalamus and the putative ability of AGEs to promote hypothalamic inflammation and dysfunction.

We appreciate the excellent suggestion but, unfortunately, this was not one of the objectives of the study, which evaluated AGEs in several tissues, so we didn’t collect the hypothalamus of the animal models.

Reviewer 2 Report

Brief summary

The aim of this study was to study the role of fruit juices sugars in inducing overweight, hyperglycemia, glycation and oxidative stress in normal and diabetic animal models.

Findings

Sugary solutions impaired energy balance regulation, leading to higher caloric intake than ad libitum fruit juices and controls, as well as weight gain, fasting hyperglycaemia, insulin intolerance and impaired oxidative stress/glycation markers in several tissues.

Strengths

Added sugars have a completely different effect in the modulation of satiety and regulation of energy balance, leading to a poorer glycemic profile and increased levels of glycation and oxidative stress markers, particularly in tissues like the heart and the kidney. The study will help to create awareness for the need of better food policies and nutritional advices for the general population.

Minor issues

There is no limitations and conclusions section. Please add them to enhance the understanding of the study by the readers and the clarity of the manuscript.

There are some typographic errors. There are very long sentences. To improve readability, consider breaking them into multiple sentences. The authors are encouraged to proof-read thoroughly the text before resubmission. English must be excellent.

Author Response

Brief summary: The aim of this study was to study the role of fruit juices sugars in inducing overweight, hyperglycemia, glycation and oxidative stress in normal and diabetic animal models.

Findings: Sugary solutions impaired energy balance regulation, leading to higher caloric intake than ad libitum fruit juices and controls, as well as weight gain, fasting hyperglycaemia, insulin intolerance and impaired oxidative stress/glycation markers in several tissues.

Strengths: Added sugars have a completely different effect in the modulation of satiety and regulation of energy balance, leading to a poorer glycemic profile and increased levels of glycation and oxidative stress markers, particularly in tissues like the heart and the kidney. The study will help to create awareness for the need of better food policies and nutritional advices for the general population.

We are grateful to the reviewer for the positive evaluation of our manuscript. Please find below a point-by-point response to all the comments, questions and suggestions raised.

Minor issues

There is no limitations and conclusions section. Please add them to enhance the understanding of the study by the readers and the clarity of the manuscript.

Thanks for the suggestion, we have included such information

There are some typographic errors. There are very long sentences. To improve readability, consider breaking them into multiple sentences. The authors are encouraged to proof-read thoroughly the text before resubmission. English must be excellent.

The manuscript was carefully revised.

Round 2

Reviewer 1 Report

Nevertheless, I have some concerns relative to this study. In particular, the rational of using two separate rat models is not defined. It is not clear why wild-type rats were not fed the sugary drink (added sugar)? It looks like that the experiments involving GK rats were “complete” including all three different interventions: water (control), fruit juices and sugary solutions. However, the same is not true for Wistar rats.

Thank you for the considerations. We have used two separate models because the objective of the study was to compare fruit juices with sugary solutions in a diabetic model. Wistar rats were used only as controls. In fact, we have also administered fruit juices to normal rats because our initial hypothesis was that fruit juices would have some harmful effects in diabetic rats and Wistar rats treated with fruit juices would be the control groups for that. In fact, the effects of the fruit juices in diabetic rats were scarce but we opted by maintaining the Wistar-fruit juices groups in the manuscript. However, sugary solutions were only administered to diabetic rats because it was the main objective of the study. Please notice that studying all samples in Wistar rats would lead to eight more experimental groups, which would create a very confuse manuscript and decrease the focus of the study.

In the reviewer’s opinion, the lack of the remaining groups, rather than their inclusion, generate confusion. What is the effect of a sugary solution in the Wistar rats? Whether the sugary solution would induce detrimental effects only in the diabetic rats remains to be elucidated. The effects on GK and Wistar could have been separated to avoid any potential confusion.

The experiment for the quantification of AGEs is not well balanced. While for the fruit juice the formation of AGEs was only dependent on endogenous protein content, in the sugary drinks there was an addition of BSA. It would be important to match the amount of BSA in the juices and sugary drinks to confirm that the increased formation of AGEs in the sugary drinks is not dependent on BSA.

The BSA used was the same present in the fruit juice of each sample to match the total protein content. Such information was present in the first version of the manuscript “The concentration of BSA used was matched with the concentration of total protein in each of the juice samples”, in section 2.3

I completely understand that BSA was used to match protein content between fruit juices and sugary drinks. However, by the sound of it, it looks like BSA was only added in the sugary drinks, but not in the fruit juices: “Matched sugary solutions were incubated in the same conditions, in the presence or not of BSA. The concentration of BSA used was matched with the concentration of total protein in each of the juice samples”. How can you be sure that the difference in AGE formation is not due to the fact that BSA was only added to sugary drinks (independently of total amount of proteins)?

Figures are not easy to follow as the graphical quality, at least with regard to the text provided to the reviewer, is very low. Es. Figure 2 B-I. This is particularly evident for x-y graphs.

The font size was increased in order to improve readability

Figures are still difficult to follow.

What is the rationale of: Comparing Wistar and GK rats?

Wistar rats were used as the control situation for the diabetic model.

Wistar rats remain a control only with regard to fruit juices. This aspect should be emphasised in the discussion.

Figure 9: Was the staining quantified?

No, we opted by not quantifying the staining because we used order quantitative or semi-quantitative markers, namely AGEs and antioxidant enzymes. Quantifying fluorescence may produce biased results so we opted for only showing the images obtained.

To avoid potential bias, fluorescence could have been quantified by a blinded researcher not involved in the study.

In terms of the regulation of energy intake and energy balance, another important aspect to consider is the role of the hypothalamus and the putative ability of AGEs to promote hypothalamic inflammation and dysfunction.

We appreciate the excellent suggestion but, unfortunately, this was not one of the objectives of the study, which evaluated AGEs in several tissues, so we didn’t collect the hypothalamus of the animal models.

I am aware the hypothalamus was not analysed. However, this aspect could be included in the discussion as a potential mechanism underpinning the effects of sugary drinks.

Author Response

We thank the reviewer for the comments and we look forward to improve the manuscript accordingly. Find below point-by-point responses (in green text) to each comment (in black text) and previous answer (blue text).

Nevertheless, I have some concerns relative to this study. In particular, the rational of using two separate rat models is not defined. It is not clear why wild-type rats were not fed the sugary drink (added sugar)? It looks like that the experiments involving GK rats were “complete” including all three different interventions: water (control), fruit juices and sugary solutions. However, the same is not true for Wistar rats.

Thank you for the considerations. We have used two separate models because the objective of the study was to compare fruit juices with sugary solutions in a diabetic model. Wistar rats were used only as controls. In fact, we have also administered fruit juices to normal rats because our initial hypothesis was that fruit juices would have some harmful effects in diabetic rats and Wistar rats treated with fruit juices would be the control groups for that. In fact, the effects of the fruit juices in diabetic rats were scarce but we opted by maintaining the Wistar-fruit juices groups in the manuscript. However, sugary solutions were only administered to diabetic rats because it was the main objective of the study. Please notice that studying all samples in Wistar rats would lead to eight more experimental groups, which would create a very confuse manuscript and decrease the focus of the study.

In the reviewer’s opinion, the lack of the remaining groups, rather than their inclusion, generate confusion. What is the effect of a sugary solution in the Wistar rats? Whether the sugary solution would induce detrimental effects only in the diabetic rats remains to be elucidated. The effects on GK and Wistar could have been separated to avoid any potential confusion.

We understand the opinion of the reviewer, but as stated in the first revision, that was not the objective of the study and those experiments were not performed. Please notice that the manuscript includes experimental data of 18 groups of rats. Including Wistar rats with sugary solutions would make a total of 26 groups of rats. This would make the data impossible to show in the same document.  

As we have already explained, Wistar rats treated with juices were the control group for the effects of each juice samples in GK rats. In our opinion, they must be maintained in the manuscript. They could be compared only with GK-juices in specific graphs in order to facilitate the understanding of the results, having graphs with “Wistar; Wistar-juice; GK; GK-juice”, for each sample. However, this would duplicate the number of graphs of the manuscript, because we would still need another graph to compare juices with sugary solutions in GK rats. Thus, we opted by using graphs with all the experimental groups of each sample, because having more than one graph for each parameter would be much worse to follow, in our opinion. Given that amount of data generated, we opted by having the lower number of graphs possible, to avoid confusion. In one of our first drafts, we had some figures using such layout and it was abandoned because it would be impossible to include everything in the manuscript.

The experiment for the quantification of AGEs is not well balanced. While for the fruit juice the formation of AGEs was only dependent on endogenous protein content, in the sugary drinks there was an addition of BSA. It would be important to match the amount of BSA in the juices and sugary drinks to confirm that the increased formation of AGEs in the sugary drinks is not dependent on BSA.

The BSA used was the same present in the fruit juice of each sample to match the total protein content. Such information was present in the first version of the manuscript “The concentration of BSA used was matched with the concentration of total protein in each of the juice samples”, in section 2.3

I completely understand that BSA was used to match protein content between fruit juices and sugary drinks. However, by the sound of it, it looks like BSA was only added in the sugary drinks, but not in the fruit juices: “Matched sugary solutions were incubated in the same conditions, in the presence or not of BSA. The concentration of BSA used was matched with the concentration of total protein in each of the juice samples”. How can you be sure that the difference in AGE formation is not due to the fact that BSA was only added to sugary drinks (independently of total amount of proteins)?

Indeed, BSA was only added to sugary solutions, because it would increase the protein content of juices and create an experimental error, biasing the results. The formation of AGEs does not depend of the specific protein, because sugars and carbonyls do not react with specific proteins, but with specific residues of all proteins like arginine or lysine. So, we used a standard protein to match the total amount of protein present in each of the juice samples.

https://pubmed.ncbi.nlm.nih.gov/27636890/

https://pubmed.ncbi.nlm.nih.gov/31539311/

Figures are not easy to follow as the graphical quality, at least with regard to the text provided to the reviewer, is very low. Es. Figure 2 B-I. This is particularly evident for x-y graphs.

The font size was increased in order to improve readability

Figures are still difficult to follow.

Please notice that in R1 version we tried to keep the figure and the respective legend at the same page, but in the end is the production team from the journal who decides that, so we cannot control the size of the figures and we believe the font size in enough for a pdf view. More, we have used different colours for each sample to improve understandability. We have further increased the size of the figures, but at the end that will not be authors decision.

What is the rationale of: Comparing Wistar and GK rats?

Wistar rats were used as the control situation for the diabetic model.

Wistar rats remain a control only with regard to fruit juices. This aspect should be emphasised in the discussion.

Wistar rats are controls for Wistar-juices groups and also for the diabetic phenotype of GK rats. We believe we should compare the effects of juices and sugary solutions in a specific parameter of GK rats knowing what is the normal phenotype. In many of the parameters studied GK rats have non-significant alterations when comparing with the normal rats, which are further aggravated mainly by the sugary solutions. That is an extremely important conclusion that cannot be underestimated. We have included information regarding this in the discussion.

Figure 9: Was the staining quantified?

No, we opted by not quantifying the staining because we used order quantitative or semi-quantitative markers, namely AGEs and antioxidant enzymes. Quantifying fluorescence may produce biased results so we opted for only showing the images obtained.

To avoid potential bias, fluorescence could have been quantified by a blinded researcher not involved in the study.

The fluorescence was quantified and the graphs were included in figure 9

In terms of the regulation of energy intake and energy balance, another important aspect to consider is the role of the hypothalamus and the putative ability of AGEs to promote hypothalamic inflammation and dysfunction.

We appreciate the excellent suggestion but, unfortunately, this was not one of the objectives of the study, which evaluated AGEs in several tissues, so we didn’t collect the hypothalamus of the animal models.

I am aware the hypothalamus was not analysed. However, this aspect could be included in the discussion as a potential mechanism underpinning the effects of sugary drinks.

We have included such information in the discussion
